# AN EMPIRICAL STUDY OF THE EXPRESSIVENESS OF GRAPH KERNELS AND GRAPH NEURAL NETWORKS

## ABSTRACT

Graph neural networks and graph kernels have achieved great success in solving machine learning problems on graphs. Recently, there has been considerable interest in determining the expressive power mainly of graph neural networks and of graph kernels, to a lesser extent. Most studies have focused on the ability of these approaches to distinguish non-isomorphic graphs or to identify specific graph properties. However, there is often a need for algorithms whose produced graph representations can accurately capture similarity/distance of graphs. This paper studies the expressive power of graph neural networks and graph kernels from an empirical perspective. Specifically, we compare the graph representations and similarities produced by these algorithms against those generated by a well-accepted, but intractable graph similarity function. We also investigate the impact of node attributes on the performance of the different models and kernels. Our results reveal interesting findings. For instance, we find that theoretically more powerful models do not necessarily yield higher-quality representations, while graph kernels are shown to be very competitive with graph neural networks.

## 1 INTRODUCTION

In recent years, graph-structured data has experienced an enormous growth in many domains, ranging from chemo- and bio-informatics to social network analysis. Several problems of increasing interest require applying machine learning techniques to graph-structured data. Examples of such problems include predicting the quantum mechanical properties of molecules (Gilmer et al., 2017) and modeling physical systems (Battaglia et al., 2016). To develop successful machine learning models in the domain of graphs, we need techniques that can both extract the information that is hidden in the graph structure, and also exploit the information contained within node and edge attributes. In the past years, the problem of machine learning on graphs has been governed by two major families of approaches, namely graph kernels (Nikolentzos et al., 2019) and graph neural networks (GNNs) (Wu et al., 2020).

Recently, much research has focused on measuring the expressive power of GNNs (Xu et al., 2019; Morris et al., 2019; Murphy et al., 2019; Maron et al., 2019a;b; Sato et al., 2019; Keriven & Peyré, 2019; Chen et al., 2019; Dasoulas et al., 2020; Nikolentzos et al., 2020; Barceló et al., 2020). On the other hand, in the case of graph kernels, there was a limited number of similar studies (Kriege et al., 2018). This is mainly due to the fact that the landscape of graph kernels is much more diverse than that of GNNs. Indeed, although numerous GNN variants have been recently proposed, most of them share the same basic idea, and can be reformulated into a single common framework, so-called message passing neural networks (MPNNs) (Gilmer et al., 2017). These models employ a message passing procedure to aggregate local information of vertices and are closely related to the Weisfeiler-Lehman (WL) test of graph isomorphism, a powerful heuristic which can successfully test isomorphism for a broad class of graphs (Arvind et al., 2015).

When dealing with learning problems on graphs, a practitioner needs to choose one GNN or one graph kernel for her particular application. The practitioner is then faced with the following question: Does this GNN variant or graph kernel capture graph similarity better than others? Unfortunately, this question is far from being answered. Most of the above studies investigate the power of GNNs in terms of distinguishing between non-isomorphic graphs or in terms of how well they can approximate combinatorial problems. However, in graph classification/regression problems, we are

not that much interested in testing whether two (sub)graphs are isomorphic to each other, but mainly in classifying graphs or in predicting real values associated with these graphs. In such tasks, it has been observed that stronger GNNs do not necessarily outperform weaker GNNs. Therefore, it seems that the design of GNNs is driven by theoretical considerations which are not realistic in practical settings. Ideally, we would like to learn representations which accurately capture the similarities or distances between graphs.

A practitioner can then choose an algorithm based on its empirical performance. Indeed, GNNs and graph kernels are usually evaluated on standard datasets derived from bio-/chemo-informatics and from social media (Morris et al., 2020). However, several concerns have been raised recently with regards to the reliability of those datasets, mainly due to their small size and to inherent isomorphism bias problems (Ivanov et al., 2019). More importantly, it has been observed that the adopted experimental settings are in many cases ambiguous or not reproducible (Errica et al., 2020). The experimental setup is not standardized across different works, and there are often many issues related to hyperparameter tuning and to how model selection and model assessment are performed. These issues easily generate doubts and confusion among practitioners that need a fully transparent and reproducible experimental setting.

**Present work.** In this paper, we empirically evaluate the expressive power of GNNs and graph kernels. Specifically, we build a dataset that contains instances of different families of graphs. Then, we compare the graph representations and similarities produced by GNNs and graph kernels against those generated by an intractable graph similarity function which we consider to be an oracle function that outputs the true similarity between graphs. We perform a large number of experiments where we compare several different kernels, architectures, and pooling functions. Secondly, we study the impact of node attributes on the performance of the different models and kernels. We show that annotating the nodes with their degree and/or triangle participation can be beneficial in terms of performance in the case of GNNs, while it is not very useful in the case of graph kernels. Finally, we investigate which pairs of graphs (from our dataset) lead GNNs and kernels to the highest error in the estimated simialrity. Surprisingly, we find that several GNNs and kernels assign identical or similar representations to very dissimilar graphs. We publicly release code and dataset to reproduce our results, in order to allow other researchers to conduct similar studies[1].

## 2 RELATED WORK

Over the past years, the expessiveness of graph kernels was assessed almost exclusively from experimental studies. Therefore, still, there is no theoretical justification on why certain graph kernels perform better than others. From the early days of the field, it was clear though that the mapping induced by kernels that are computable in polynomial time is not injective (and thus these kernels cannot solve the graph isomorphism problem) (Gärtner et al., 2003). Recently, Kriege et al. (2018) proposed a framework to measure the expressiveness of graph kernels based on ideas from property testing. It was shown that some well-established graph kernels such as the shortest path kernel, the graphlet kernel, and the Weisfeiler-Lehman subtree kernel cannot identify basic graph properties such as planarity or bipartitness.

With the exception of the work of Scarselli et al. (2008), until recently, there has been little attempt to understand the expressive power of GNNs. Several recent studies have investigated the connections between GNNs and the Weisfeiler-Lehman (WL) test of isomorphism and its higher-order variants. For instance, it was shown that standard GNNs do not have more power in terms of distinguishing between non-isomorphic graphs than the WL algorithm (Morris et al., 2019; Xu et al., 2018). Morris et al. (2019) proposed a family of GNNs which rely on a message passing scheme between subgraphs of cardinality $k$, and which have exactly the same power in terms of distinguishing non-isomorphic graphs as the set-based variant of the $k$-WL algorithm. In a similar spirit, (Maron et al., 2019a) introduced $k$-order graph networks which are at least as powerful as the folklore variant of the $k$-WL graph isomorphism test in terms of distinguishing non-isomorphic graphs. These models were also shown to be universal (Maron et al., 2019c; Keriven & Peyré, 2019), but require using high order tensors and therefore are not practical. Chen et al. (2019) show that the two main approaches for studying the expressive power of GNNS, namely graph isomorphism testing and invariant function

---

[1] Code available at: `https://github.com/xxxxx/xxxxx/`

approximation, are equivalent to each other. Futhermore, the authors propose a GNN that is more powerful than 2-WL. Barceló et al. (2020) rely on a connection between the WL algorithm and first order logic, and characterize the expressive power of GNNs in terms of classical logical languages. The impact of random features on the expressive power of MPNNs is considered in Sato et al. (2020). Nikolentzos et al. (2020) show that standard GNNs cannot identify fundamental graph properties such as triangle-freeness and connectivity, and they propose a model that can identify these properties. Murphy et al. (2019) and Dasoulas et al. (2020) take into account all possible node permutations and produce universal graph representations. The emerging problems becomes intractable once the number of nodes is large and they propose approximation schemes to make the computation tractable in practice. Some recent works have studied the generalization properties of GNNs (Scarselli et al., 2018; Verma & Zhang, 2019; Garg et al., 2020). For a comprehensive overview of the expressive power of GNNs, the interested reader is referred to Sato (2020).

## 3 COMPARING GRAPHS TO EACH OTHER

Formally, for any two graphs $G_1 = (V_1, E_1)$ and $G_2 = (V_2, E_2)$ on $n$ vertices with respective $n \times n$ adjacency matrices $\boldsymbol{A}_1$ and $\boldsymbol{A}_2$, we define a function $f : \mathcal{G} \times \mathcal{G} \to \mathbb{R}$ where $\mathcal{G}$ is the space of graphs which quantifies the similarity of $G_1$ and $G_2$. Note that in the literature, this problem is often referred to as *graph comparison*. The problem of graph comparison has been the topic of numerous studies in computer science (Conte et al., 2004). Existing approaches are generally motivated by runtime considerations. These approaches are computable in polynomial time, but also limited in terms of expressive power. For instance, for similarity measures that take values between $0$ and $1$, $f(G_1, G_1) = f(G_1, G_2) = 1$ may hold even if $G_1$ and $G_2$ are not isomorphic to each other. In this paper, we consider a graph comparison function which is not computable in polynomial time. The function can be expressed as a maximization problem, and is defined as follows:

$$f(G_1, G_2) = \max_{\boldsymbol{P} \in \Pi} \frac{\sum_{i=1}^{n} \sum_{j=1}^{n} \left[\boldsymbol{A}_1 \odot \boldsymbol{P} \boldsymbol{A}_2 \boldsymbol{P}^{\top}\right]_{ij}}{||\boldsymbol{A}_1||_F \, ||\boldsymbol{A}_2||_F} \tag{1}$$

where $\Pi$ denotes the set of $n \times n$ permutation matrices, $\odot$ denotes the elementwise product, and $|| \cdot ||_F$ is the Froebenius matrix norm. For clarity of presentation we assume $n$ to be fixed (i. e., both graphs consist of $n$ vertices). In order to apply the function to graphs of different cardinality, one can append zero rows and columns to the adjacency matrix of the smaller graph to make its number of rows and columns equal to $n$. Therefore, the problem of graph comparison can be reformulated as the problem of maximizing the above function over the set of permutation matrices. A permutation matrix $\boldsymbol{P}$ gives rise to a bijection $\pi : V_1 \to V_2$. The function defined above seeks for a bijection such that the number of common edges $|\{(u, v) \in E_1 : (\pi(u), \pi(v)) \in E_2\}|$ is maximized. Then, the number of common edges is normalized such that it takes values between $0$ and $1$. Observe that the above definition is symmetric in $G_1$ and $G_2$. The two graphs are isomorphic to each other if and only if there exists a permutation matrix $\boldsymbol{P}$ for which the above function is equal to $1$. Therefore, a value equal to $0$ denotes maximal dissimilarity, while a value equal to $1$ denotes that the two graphs are isomorphic to each other. Note that if the compared graphs are not empty (i. e., they contain at least one edge), the function will take some value greater than $0$. Solving the above optimization problem for large graphs is clearly intractable since there are $n!$ permutation matrices of size $n$. As mentioned above, existing approaches for comparing graphs to each other or for generating graph representations are generally motivated by runtime considerations, and provide no guarantees on how well they can approximate the above function or other similar graph comparison functions. In this paper, we investigate how different graph comparison/representation learning approaches approximate the above-defined function from an empirical standpoint.

## 4 EMPIRICAL EVALUATION

### 4.1 DATASET

Since the function defined in equation 1 is intractable for large graphs, we generated graphs consisting of at most 9 vertices. Furthermore, each graph is connected and contains at least 1 edge. We generated 191 pairwise non-isomorphic graphs. The dataset consists of different types of synthetic graphs. These include simple structures such as cycle graphs, path graphs, grid graphs, complete

| Synthetic Dataset | |
|---|---|
| Max # vertices | 9 |
| Min # vertices | 2 |
| Average # vertices | 7.29 |
| Max # edges | 36 |
| Min # edges | 1 |
| Average # edges | 11.34 |
| # graphs | 191 |

Table 1: Summary of the synthetic dataset that we used in our experiments.

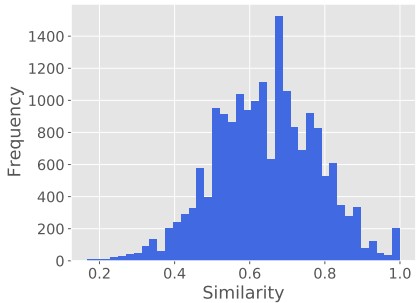

Figure 1: Distribution of similarities between the synthetically generated graphs.

graphs and star graphs, but also randomly-generated graphs such as Erdős-Rényi graphs, Barabási-Albert graphs and Watts-Strogatz graphs. Table 1 shows statistics of the synthetic dataset that we used in our experiments. Figure 1 illustrates the distribution of the similarities of the generated graphs as computed by the proposed measure. There are $191*192/2 = 18,336$ pairs of graphs in total (including pairs consisting of a graph and itself). Interestingly, most of the similarities take values between $0.5$ and $0.8$.

## 4.2 SELECTED APPROACHES

Suitable graph kernels and GNNs were selected according to the following criteria: (1) publicly available implementations, (2) strong architectural differences, (3) popularity, and (4) peer reviewed. We next present the graph kernels and GNNs that were included into our evaluation. For a detailed description of each kernel and each GNN, we refer the reader to their respective papers.

**Graph kernels.** We selected the following 6 graph kernels: (1) the random walk kernel (RW) (Gärtner et al., 2003), (2) the shortest path kernel (SP) (Borgwardt & Kriegel, 2005), (3) the graphlet kernel (GR) (Shervashidze et al., 2009), (4) the Weisfeiler-Lehman subtree kernel (WL) (Shervashidze et al., 2011), (5) the pyramid match kernel (PM) (Nikolentzos et al., 2017), and (6) the GraphHopper kernel (Feragen et al., 2013). Note that GR can only operate on unlabeled graphs. The rest of the kernels can handle graphs with discrete node labels, while GraphHopper is the only kernel that can deal with continuous multi-dimensional node features.

**GNNs.** We also selected the following GNNs: (1) GCN (Kipf & Welling, 2016), (2) GAT (Veličković et al., 2017), (3) 1-GNN (Morris et al., 2019), (4) GG-NN (Li et al., 2015), (5) Graph-SAGE (Hamilton et al., 2017), (6) GIN (Xu et al., 2019), (7) ChebNet (Defferrard et al., 2016), (8) ARMA (Bianchi et al., 2019), (9) 1-2-GNN (Morris et al., 2019), (10) $k$-hop GNN (Nikolentzos et al., 2020), and (11) Provably Powerful GNN (Maron et al., 2019a). To produce graph representations, we use 3 common pooling functions, namely the sum aggregator, the mean aggregator and the max aggregator.

## 4.3 BASELINES

We utilize some simple baselines whose purpose is to understand the extent to which graph kernels and GNNs can indeed learn representations that capture graph similarity. The first baseline is a function that randomly computes the similarity of two graphs by sampling a number from $[0, 1]$ with uniform probability. The second baseline is a constant function. The outputs of this functions is equal to $1$. Using such simple baselines as a reference is crucial since they can provide feedback on the effectiveness of graph kernels and GNNs in the considered task. If the performance of a GNN or kernel is close to that of one of the baselines, that would mean that the GNN/kernel fails to encode accurately graph representations, and therefore, graph similarities.

## 4.4 EXPERIMENTAL SETTINGS

**Normalization.** As discussed above, the function defined in equation 1 gives an output in the range $[0, 1]$. The similarity of two graphs $G_1$ and $G_2$ is equal to $1$ if and only if $G_1$ and $G_2$ are isomorphic

to each other. We normalize the obtained kernel values as follows such that they also take values in the range $[0, 1]$: $k_{ij} = k_{ij}/\sqrt{k_{ii}}\sqrt{k_{jj}}$. where $k_{ij}$ is the kernel between graphs $G_i$ and $G_j$. For the GNNs, we compute the cosine similarity between the graph representations of the penultimate layer as follows: $z_i^\top z_j / ||z_i|| \, ||z_j||$ where $z_i$ is the representation of graph $G_i$. Note that the ReLU function has already been applied to these representations, and thus the cosine similarity also takes values between $0$ and $1$. In fact, the two employed normalization schemes (i.e., for kernels and GNNs) are equivalent since a kernel value corresponds to an inner product between the representations of two graphs in some Hilbert space.

We should stress that the normalized kernel value and cosine similarity can take a value equal to $1$ even if the compared graphs are mapped to different representations. Indeed, if the angle between the vector representations of two graphs is $0°$, then their normalized kernel value/cosine similarity is equal to $1$. To avoid such a scenario, we could define a distance function between graphs and accordingly compute the Euclidean distance between the graph representations generated by the different approaches. However, it turns out that most widely-used learning algorithms compute the inner products between the input objects or some transformations of these objects. In fact, when learning with kernels, we usually normalize the kernel matrices using the equation defined above before feeding to a kernel method such as the SVM classifier. Therefore, we believe that evaluating the "similarity" of the obtained representations is of higher significance than evaluating their "distance".

**Evaluation metrics.** To assess how well the different approaches approximate the similarity function defined in equation 1, we employed two evaluation metrics: the Pearson correlation coefficient and the mean squared error (MSE). The Pearson correlation coefficient measures the linear relationship between two variables. It takes values between $-1$ and $1$, while a value of $1$ denotes total positive linear correlation. In our setting, a high value of correlation would mean that the approach under consideration captures the relationships between the similarities (e.g., whether the similarity of a pair of graphs is greater or lower than that of another pair). The second measure, MSE, is equal to the average squared difference between the estimated values and the actual values. A very small value of MSE denotes that the derived similarities are very close to those produced by the function defined in equation 1. A credible graph representation learning/similarity approach would yield both a high correlation and a small MSE. The former would ensure that similar/dissimilar graphs are indeed deemed similar/dissimilar by the considered approach, while the latter would verify that the similarity values are on par with those produced by the function defined in equation 1. Note that the correlation between the output of a constant function and the output of equation 1 is not defined since the values produced by the constant function have a variance equal to zero.

**Hyperparameters.** For RW, we set $\lambda$ to $0.01$. For GR, we count all the graphlets of size 3. For the WL kernel, we choose the number of iterations from $\{1, 2, \ldots, 6\}$. For PM, the dimensionality of the embeddings $d$ and the number of levels $L$ are set equal to $6$ and $4$, respectively. For the GraphHopper kernel, a linear kernel is used on the continuous-valued attributes.

For the GNNs, we use 2 neighborhood aggregation layers. For GraphSAGE, we use the mean function to aggregate the neighbors of a node. For ChebNet, we use polynomials of order 2, and for ARMA, we set the number of stacks $K$ to 2 and the depth $T$ also to 2. The hidden-dimension size of the neighborhood aggregation layers is set equal to $64$. We apply the ReLU activation function to the output of each neighborhood aggregation layer. As mentioned above, we use 3 common readout functions: sum, mean and max. The output of the readout function is passed on to a fully-connected layer with 32 hidden units followed by ReLU nonlinearity. The output of the ReLU function corresponds to the representation of the input graph (i.e., each graph is represented by a 32-dimensional vector). For Provably Powerful GNN, we use a network suffix that consists of an invariant readout function followed by 2 fully connected layers. To train all neural networks, we use the Adam optimizer with learning rate $0.001$. We set the batch size to 32 and the number of epochs to $100$. We store the model that achieved the best validation accuracy into disk. At the end of training, the model is retrieved from the disk, and we use it to generate graph representations.

**Protocol.** For deterministic approaches, we compute the graph similarities once and report the emerging correlation and MSE. For the remaining approaches (e.g., GNNs, since their parameters are randomly initialized), we repeat the experiment 10 times and report the average correlation, the average MSE and the corresponding standard deviations.

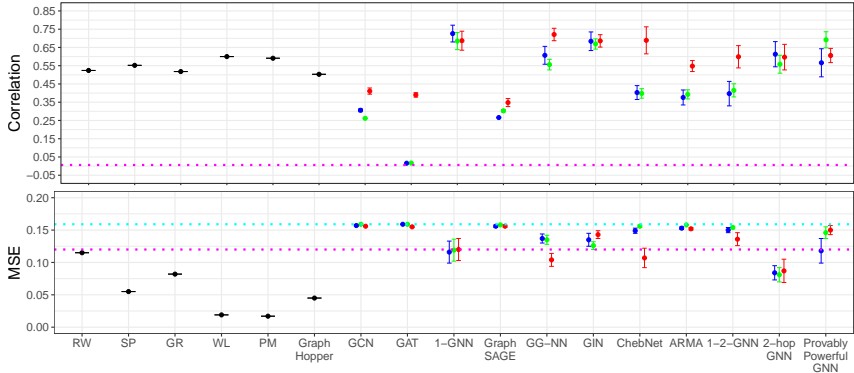

Figure 2: Performance of the different approaches without node features. For GNNs, the different colors indicate the three pooling functions: sum (●), mean (●), and max (●). The horizontal lines correspond to the two baselines (random, constant).

**Implementations.** For the graph kernels, we used the implementations contained in the GraKeL library (Siglidis et al., 2020). For 1-2 GNN, $k$-hop GNN, and Provably Poewrful GNN, we use the publicly available implementations provided by the authors. The remaining GNNs were implemented using the Pytorch Geometric library (Fey & Lenssen, 2019).

## 4.5 RESULTS

### 4.5.1 NO FEATURES

In the first set of experiments, we do not use any pre-computed features, and therefore, the emerging representations/similarities rely solely on the representational capabilities of the different approaches. Note that all GNNs except Provably Powerful GNN and most kernels require the nodes to be annotated with either continuous attributes or discrete labels. Therefore, for these approaches, we annotate each node with a single feature (i. e., the discrete label 1 or the real value 1.0).

For GNNs, we consider two alternatives: (1) we randomly initialize the parameters of the models and we perform a feedforward pass to generate the graph representations or (2) we randomly initialize the parameters of the models, we train the models on some independent dataset and then we perform the feedforward pass to generate the representations. Kernels are, in a sense, unsupervised, and cannot be trained on any dataset. The obtained results for the two aforementioned cases are illustrated in Figures 2 and 3. Note that in the second case, the models were trained on the IMDB-BINARY graph classification dataset (results for two other datasets are given in the Appendix).

In terms of correlation, 1-GNN and GIN are the best-performing approaches followed by GG-NN and one variant of Chebnet. The 6 kernels along with 2-hop GNN and Provably Powerful GNN perform slighly worse than the above models. In terms of MSE, the kernels outperform in general the GNNs. Notably, some kernels such as WL and PM achieve very low values of MSE. On the other hand, most GNNs fail to outperform the random baseline. Specifically, the only models that outperform this baseline are 1-GNN, 2-hop GNN, GG-NN with sum pooling, Chebnet with sum pooling and Provably Powerful GNN with max pooling. With regards to the three pooling functions, the sum operator seems to outperform the others. Furthermore, it is important to mention that more powerful architectures (e. g., 1-2-GNN, 2-hop GNN, Provably Powerful GNN) do not necessarily lead to higher correlation and lower MSE than standard, less expressive GNNs. We next investigate how the performance of the GNNs changes when the models are first trained on the IMDB-BINARY dataset. The results are shown in Figure 3. We observe that there is a decrease in correlation, and no GNN achieves higher correlation than the ones achieved by the WL and PM kernels anymore. On the other hand, the MSE of most GNNs also decreases. For instance, most GNNs now yield lower MSEs than the random baseline. However, still, GCN, GAT and GraphSAGE fail to outperform this baseline.

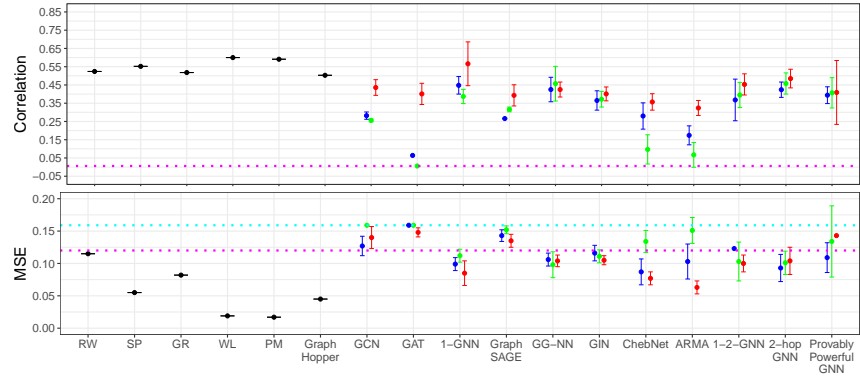

Figure 3: Performance of the different approaches without node features. The GNNs were trained on the IMDB-BINARY dataset, and the different colors indicate the three pooling functions: sum (●), mean (●), and max (●). The horizontal lines correspond to the two baselines (random, constant).

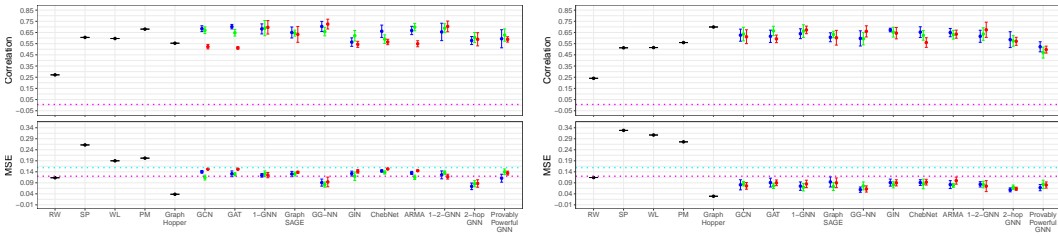

Figure 4: Performance of the different approaches with node features. For GNNs, the different colors indicate the three pooling functions: sum (●), mean (●), and max (●). The horizontal lines correspond to the two baselines (random, constant).

## 4.6 THE EFFECT OF NODE FEATURES

In GNN literature, it is common practice to use local features (e. g., degree) as node attributes. In previous studies, it has been reported that using node degrees as input features leads to an increase in performance on almost all graph classification datasets (Errica et al., 2020). We next investigate what is the impact of such features on the learned graph representations. In the first set of experiments, we assign a single feature to each node, i. e., its degree. In the second set of experiments, each node is annotated with a 2-dimensional vector where the two elements correspond to its degree and to the number of triangles in which it participates. Note that the GR kernel cannot handle node labels/attributes, and hence, it is excluded from the evaluation. Furthemore, all the other kernels except GraphHopper can only handle discrete node labels. For the first set of experiments, since degrees are natural numbers, we can treat them as node labels. For the second set of experiments, we map each unique pair of features (i. e., degree and trangle participation) to a natural number.

Figures 4 and 5 illustrate the obtained results for the case of randomly initialized GNNs and GNNs trained on IMDB-BINARY, respectively. We observe that GraphHopper, the only kernel that can naturally handle multi-dimensional continuous node features, takes advantage of the node degree and triangle participation information since it exhibits a very high correlation and a small MSE. On the other hand, the quality of the representations learned by the remaining kernels seems to be lower when these features are taken into account. The addition of the degrees leads to slight increase in correlation, but also to a large increase in MSE. When both features are considered, there is still a large increase in MSE, but also a decrease in correlation. This suggests that for kernels that are not designed to operate on graphs with continuous node attributes, using these features may result into a decrease in performance. In the case of randomly initialized GNNs, in the first set of experiments, we observe an increase in the correlation between most models and the considered graph similarity function. The models, however, reach similar MSE levels to the previous ones, and most of them are still outperformed by the random baseline. This changes when triangle participation is taken into account. The correlations remain high, while now all the models outperform the random baseline. 1-2-GNN, 1-GNN and GG-NN achieve the highest levels of correlation, while 2-hop GNN, GG-NN and Provably Powerful GNN are the best-performing models in terms of MSE. Again, more compli-

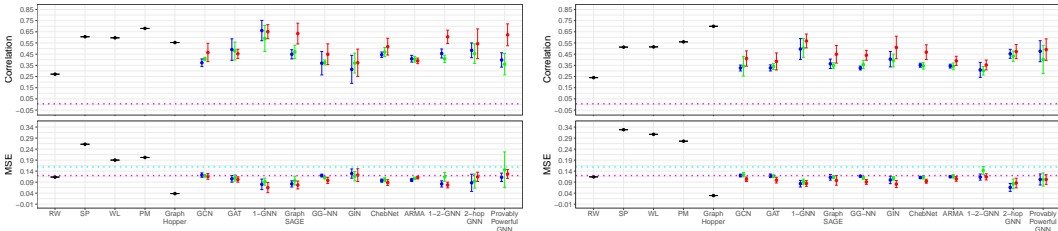

Figure 5: Performance of the different approaches with node features. The GNNs were trained on the IMDB-BINARY dataset, and the different colors indicate the three pooling functions: sum (●), mean (●), and max (●). The horizontal lines correspond to the two baselines (random, constant).

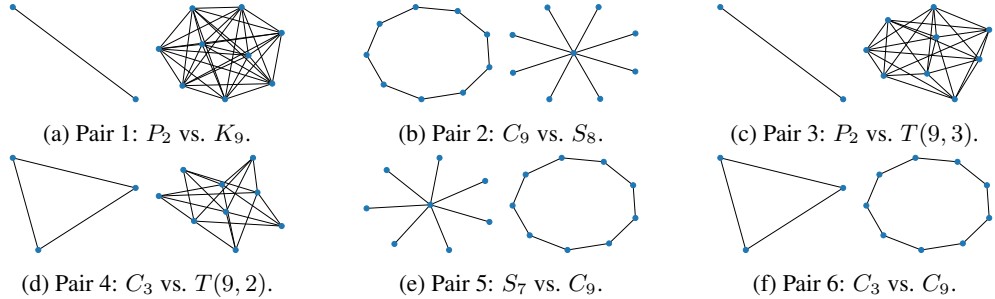

(a) Pair 1: $P_2$ vs. $K_9$.  (b) Pair 2: $C_9$ vs. $S_8$.  (c) Pair 3: $P_2$ vs. $T(9, 3)$.

(d) Pair 4: $C_3$ vs. $T(9, 2)$.  (e) Pair 5: $S_7$ vs. $C_9$.  (f) Pair 6: $C_3$ vs. $C_9$.

Figure 6: Performance of 1-GNN with with respect to the different activation functions.

cated models do not necessarily outperform simpler models. When trained on IMDB-BINARY, the correlation between the models and equation 1 decreases. At the same time, the models yield lower and slightly higher MSEs in the first and second set of experiments, respectively.

### 4.6.1 WHICH PAIRS ARE HARD?

We next find for each approach the similarity that deviates most from the one computed using equation 1. These pairs of graphs can be thought of as the most challenging for a model or kernel. For each approach, we find the pair of graphs which maximizes the absolute difference between the similarity computed using equation 1 and the one produced by the model/kernel. Note that for GNNs, we focus on the worst-case scenario independent of the pooling function. The different pairs of graphs are illustrated in Figure 6. Surprisingly, a lot of methods/kernels fail to accurately estimate the similarity of structurally very dissimilar graphs. For instance, GCN, GAT, GraphSAGE, ARMA, Chebnet, SP and GraphHopper all found the first two graphs (the path graph $P_2$ and the complete graph on 9 vertices) to be identical to each other (i. e., similarities equal to 1), while the output of equation 1 is 0.166. In the case of the two kernels, this is because the implicit vector representation of one graph is a positive scalar multiple of the representation of the other. With regards to the GNNs, note that the above models use the mean function to aggregate the representations of their neighborhoods. When such neighborhood aggregation approaches are followed by mean or max pooling functions, they produce identical representations for the $P_2$ and $K_9$ graphs. Another pair of graphs which turns out to be challenging for several approaches is the one consisting of the cycle graph with 9 vertices and the star graph $S_8$, i. e., a tree with one internal node and 8 leaves. The output of equation 1 for this pair is equal to 0.235, while the similarities produced by all the following approaches are greater than 0.900: GIN, 1-GNN, GG-NN, 2-hop GNN and RW. The next four pairs of graphs correspond to the worst-case scenarios for Provably Powerful GNN, 1-2-GNN, PM and WL, respectively. The similarity between the path graph $P_2$ and the Turán graph $T(9, 3)$ (i. e., $3^{rd}$ pair) is 0.192 according to equation 1, while the representations generated by Provably Powerful GNN yielded a similarity equal to 0.955. Likewise, the output of equation 1 is 0.258 for the $4^{th}$ pair of graphs, while 1-2-GNN produced a value equal to 0.922. The star graph $S_7$ along with the cycle graph with 9 vertices (i. e., $5^{th}$ pair) led PM to the worst similarity estimation. While equation 1 gave a similarity of 0.251, the normalized kernel value was equal to 0.773. The last pair of graphs, the cycle graph with 3 vertices and the cycle graph with 9 vertices, turns out to be the hardest for the WL kernel. While the value of equation 1 is 0.384, the normalized kernel value is equal to 1. Again, this is due to the fact that the vector representation of one graph is a positive scalar multiple of the representation of the other.

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

## A APPENDIX

### A.1 EXPANDED RELATED WORK

**Graph kernels.** Graph kernels have attracted a lot of attention in the past 20 years, and have been applied to several learning problems on graphs. A graph kernel is a symmetric, positive semidefinite function that maps graphs implicitly or explicitly into some Hilbert space, and then computes the inner product between the graph representations in that space. Early work in this area includes functions that compare substructures of graphs that are computable in polynomial time such as random walks (Gärtner et al., 2003; Kashima et al., 2003), shortest paths (Borgwardt & Kriegel, 2005) and subtrees (Ramon & Gärtner, 2003). More recently, research in the field has been directed towards more scalable approaches. This led to the development of the Weisfeiler-Lehman subtree kernel which employs a relabeling procedure inspired by the Weisfeiler-Lehman test of isomorphism (Shervashidze et al., 2011). Several extensions of the Weisfeiler-Lehman kernel have also been explored in the previous years (Morris et al., 2017; Togninalli et al., 2019; Rieck et al., 2019). Other recent works use the orthonormal representations associated with the Lovász $\vartheta$ number to produce subgraph representations (Johansson et al., 2014), employ assignment kernels to compare substructures (Kriege et al., 2016; Nikolentzos et al., 2017), compare the neighborhood subgraphs of vertices using spectral approaches (Kondor & Pan, 2016), or use randomized binning approaches (Heimann et al., 2019). Recently, some frameworks have also been proposed for deriving variants of graph kernels. These frameworks capitalize on recent advances in natural language processing (Yanardag & Vishwanathan, 2015) or on graph decomposition techniques (Nikolentzos et al., 2018a). For a thorough survey of graph kernels, the interested reader is referred to Nikolentzos et al. (2019).

**GNNs.** GNNs have recently emerged as another promising family of approaches for performing machine learning tasks on graphs. These models have attracted a lot of attention in the past few years. The main advantage of GNNs over graph kernels is that they do not require expensive Gram matrix computations and that they can naturally handle node and edge attributes (continuous values in general). Most GNN variants that have been proposed in the past years (Li et al., 2015; Defferrard et al., 2016; Kearnes et al., 2016; Lei et al., 2017; Hamilton et al., 2017; Kipf & Welling, 2017; Velickovic et al., 2018; Zhang et al., 2018) all share the same basic idea, and can be reformulated into a single common framework, known as Message Passing Neural Networks (MPNNs) (Gilmer et al., 2017). These models employ an iterative message passing procedure, where each node updates its feature vector by aggregating the feature vectors of its neighbors. This procedure is typically followed by a pooling phase where a feature vector for the entire graph is produced using some permutation invariant function. All these models correspond to minor adaptations of previous works (Sperduti & Starita, 1997; Gori et al., 2005; Scarselli et al., 2009). There have been proposed several extensions and improvements to the MPNN framework. Most studies have focused on the message passing procedure and have proposed more expressive aggregation functions (Murphy et al., 2019; Seo et al., 2019; Dasoulas et al., 2020), schemes that incorporate different local structures or high-order neighborhoods (Jin et al., 2020; Abu-El-Haija et al., 2019), approaches that utilize node positional information (You et al., 2019), non-euclidian geometry approaches (Chami et al., 2019), while others have focused on efficiency (Gallicchio & Micheli, 2020). Fewer works have focused on the pooling phase and have proposed more sophisticated pooling functions (Ying et al., 2018; Gao & Ji, 2019). Note that there exist GNNs that do not belong to the family of MPNNs (Niepert et al., 2016; Nikolentzos et al., 2018b; Bai et al., 2019). A survey of recent advancements in GNN models can be found in Wu et al. (2020).

**Expressive power of graph kernels and GNNs.** Over the past years, the expessiveness of graph kernels was assessed almost exclusively from experimental studies. Therefore, still, there is no theoretical justification on why certain graph kernels perform better than others. From the early days of the field, it was clear though that the mapping induced by kernels that are computable in polynomial time is not injective (and thus these kernels cannot solve the graph isomorphism problem) (Gärtner et al., 2003). Recently, Kriege et al. (2018) proposed a framework to measure the

expressiveness of graph kernels based on ideas from property testing. It was shown that some well-established graph kernels such as the shortest path kernel, the graphlet kernel, and the Weisfeiler-Lehman subtree kernel cannot identify basic graph properties such as planarity or bipartitness.

With the exception of the work of Scarselli et al. (2008), until recently, there has been little attempt to understand the expressive power of GNNs. Several recent studies have investigated the connections between GNNs and the Weisfeiler-Lehman (WL) test of isomorphism and its higher-order variants. For instance, it was shown that standard GNNs do not have more power in terms of distinguishing between non-isomorphic graphs than the WL algorithm (Morris et al., 2019; Xu et al., 2018). Morris et al. (2019) proposed a family of GNNs which rely on a message passing scheme between subgraphs of cardinality $k$, and which have exactly the same power in terms of distinguishing non-isomorphic graphs as the set-based variant of the $k$-WL algorithm. In a similar spirit, (Maron et al., 2019a) introduced $k$-order graph networks which are at least as powerful as the folklore variant of the $k$-WL graph isomorphism test in terms of distinguishing non-isomorphic graphs. These models were also shown to be universal (Maron et al., 2019c; Keriven & Peyré, 2019), but require using high order tensors and therefore are not practical. Chen et al. (2019) show that the two main approaches for studying the expressive power of GNNS, namely graph isomorphism testing and invariant function approximation, are equivalent to each other. Futhermore, the authors propose a GNN that is more powerful than 2-WL. Barceló et al. (2020) rely on a connection between the WL algorithm and first order logic, and characterize the expressive power of GNNs in terms of classical logical languages. The impact of random features on the expressive power of MPNNs is considered in Sato et al. (2020). Nikolentzos et al. (2020) show that standard GNNs cannot identify fundamental graph properties such as triangle-freeness and connectivity, and they propose a model that can identify these properties. Murphy et al. (2019) and Dasoulas et al. (2020) take into account all possible node permutations and produce universal graph representations. The emerging problems becomes intractable once the number of nodes is large and they propose approximation schemes to make the computation tractable in practice. Some recent works have studied the generalization properties of GNNs (Scarselli et al., 2018; Verma & Zhang, 2019; Garg et al., 2020). For a comprehensive overview of the expressive power of GNNs, the interested reader is referred to Sato (2020).

## A.2    EXPECTED VALUE OF SIMILARITY FUNCTION

The similarity function defined in equation 1 is intractable. However, we should mention that if instead of the maximum over all permutation matrices, we are interested in the expected value, this can be computed efficiently as follows:

$$\bar{f}(G_1, G_2) = \mathbb{E}_{\boldsymbol{P} \sim \Pi} \frac{\sum_{i=1}^n \sum_{j=1}^n \left[ \boldsymbol{A}_1 \odot \boldsymbol{P} \boldsymbol{A}_2 \boldsymbol{P}^\top \right]_{ij}}{\|\boldsymbol{A}_1\|_F \|\boldsymbol{A}_2\|_F} = \frac{\sum_{i=1}^n \sum_{j=1}^n \left[ \boldsymbol{A}_1 \odot \mathbb{E}_{\boldsymbol{P} \sim \Pi} \boldsymbol{P} \boldsymbol{A}_2 \boldsymbol{P}^\top \right]_{ij}}{\|\boldsymbol{A}_1\|_F \|\boldsymbol{A}_2\|_F}$$

(2)

Furthermore, note that:

$$\mathbb{E}_{\boldsymbol{P} \sim \Pi} \boldsymbol{P} \boldsymbol{A}_2 \boldsymbol{P}^\top = \frac{1}{n!} \boldsymbol{P}_1 \boldsymbol{A}_2 \boldsymbol{P}_1^\top + \frac{1}{n!} \boldsymbol{P}_2 \boldsymbol{A}_2 \boldsymbol{P}_2^\top + \ldots + \frac{1}{n!} \boldsymbol{P}_{n!} \boldsymbol{A}_2 \boldsymbol{P}_{n!}^\top = \frac{2m_2}{n(n-1)} \boldsymbol{J}_n \quad (3)$$

where $m_2$ is the number of edges of $G_2$, $n = \max(n_1, n_2)$ where $n_1$ and $n_2$ are the number of nodes of $G_1$ and $G_2$, respectively, and $J_n$ is the $n \times n$ matrix of ones. Then, from equation 2 and equation 3, we have that:

$$\bar{f}(G_1, G_2) = \frac{2m_1 \, {}^{2m_2}/n(n-1)}{\sqrt{2m_1}\sqrt{2m_2}} = \frac{2\sqrt{m_1 m_2}}{n(n-1)}$$

where $m_1$ and $m_2$ are the number of edges of $G_1$ and $G_2$, respectively. Interestingly, for classes of graphs where $\boldsymbol{A} = \boldsymbol{P}\boldsymbol{A}\boldsymbol{P}^\top$ holds for any permutation matrix $\boldsymbol{P}$, then $f(G_1, G_2) = \bar{f}(G_1, G_2)$. Clearly, the above is true for the class of complete graphs $K_n$.

## A.3    FURTHER EXPERIMETAL RESULTS

### A.3.1    IMPACT OF HYPERPARAMETERS

We next will investigate how performance varies with respect to two hyperparameters: (1) the dimensionality of the hidden layers, and (2) the number of message passing layers. The greater the

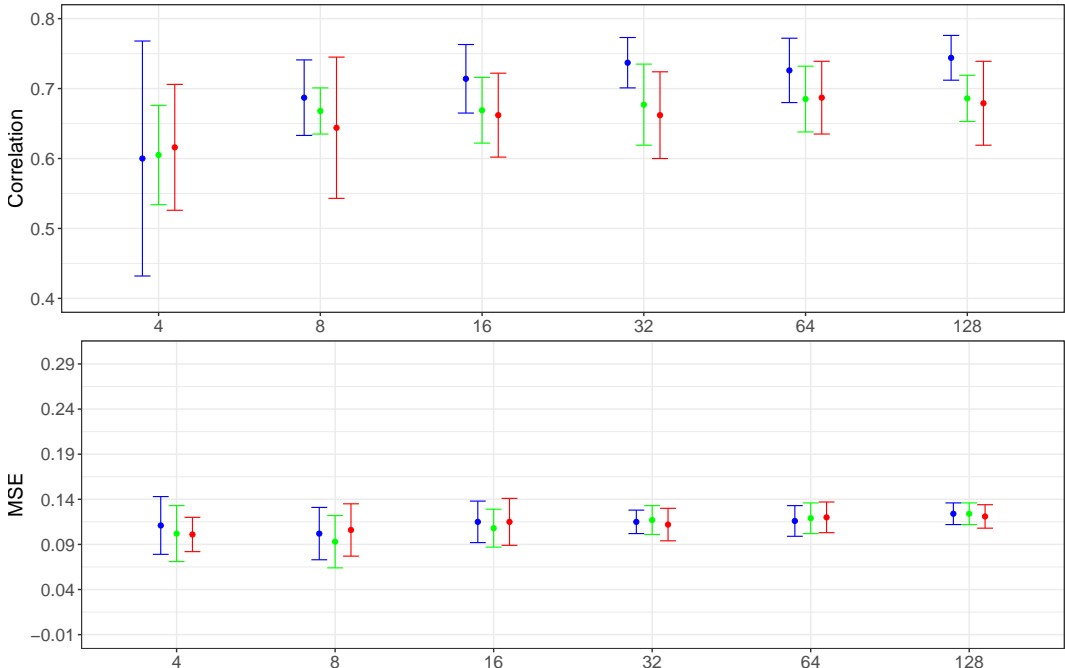

Figure 7: Performance of 1-GNN with with respect to the dimensionality of the hidden layers.

value of these two hyperparameters, the larger the capacity of the model. Therefore, we expect the models to produce better and better graph representations as these two hyperparameters increase. We study the impact of these hyperparameters on the performance of the 1-GNN model. The obtained results are shown in Figures 7 and 8. Note that the model is not trained on any dataset. We observe that the correlation between 1-GNN and the considered similarity function increases as the dimensionality of the hidden layers increases. With regards to the MSE metric, strikingly, we observe that it also slightly increases as the dimensionality of the hidden layers increases. We next investigate how performance of the 1-GNN model changes as a function of the number of message passing layers. As can be seen in Figure 8, the correlation between 1-GNN and the proposed function first increases, while then decreases. The best results are obtained for 3 message passing layers. On the other hand, MSE generally decreases as the number of message passing layers increases.

### A.3.2 IMPACT OF ACTIVATION FUNCTIONS

We also investigate what is the impact of different activation functions on the performance of 1-GNN. We experiment with four activation functions, namely tanh, sigmoid, ELU and ReLU. We also use message passing layers that contain no activation functions at all (i. e., linear activation function). Again, we do not train the model on any dataset. The obtained correlations and MSEs are given in Figure 9. We can see that ELU and tanh yield in most cases higher correlations and lower MSEs than the remaining activation functions, while sigmoid is the worst-performing activation function.

### A.3.3 GNNs TRAINED ON MUTAG AND PTC-MR

We next present some results similar to those of Figures 3 and 5. However, we now train the GNNs on two other standard graph classification datasets: MUTAG and PTC-MR. Note that these datasets contain node-labeled graphs, however, for the purposes of this study, we ignore these labels. We first present the results for the case where nodes are not annotated with any pre-computed features. The results are shown in Figures 10 and 11. We observe that GIN, 1-GNN and 1-2-GNN both trained on MUTAG and on PTC-MR generate representations that yield high values of correlation, while the representations produced by GCN and 2-hop GNN lead to the lowest levels of correlation. Interestingly, when GAT is combined with the max or mean pooling functions, its generated

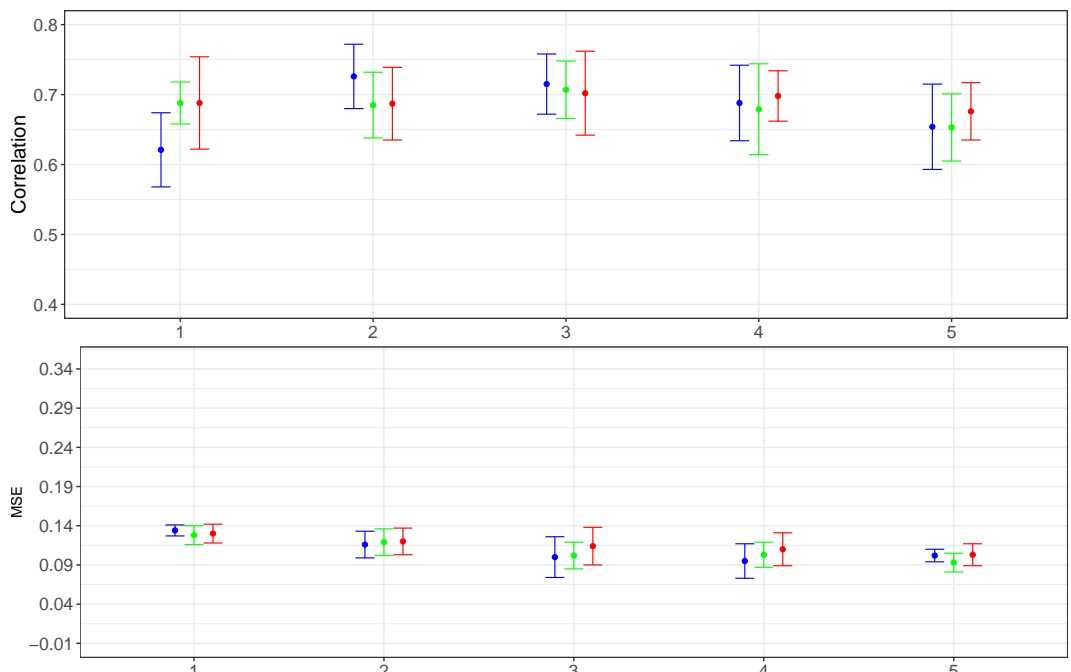

Figure 8: Performance of 1-GNN with with respect to the number of message passing layers.

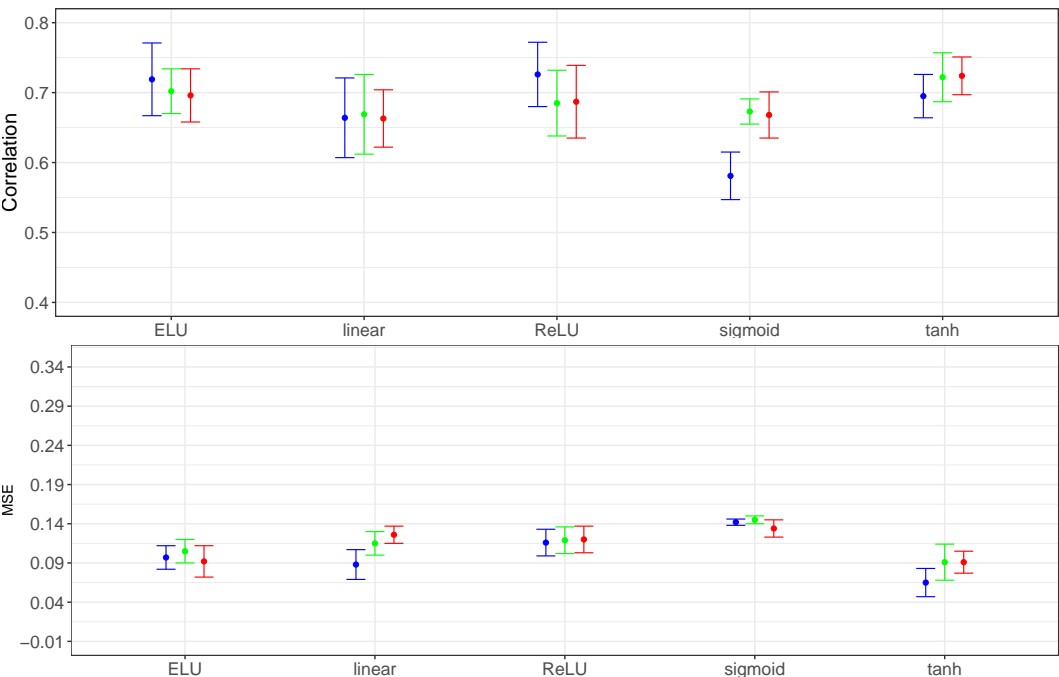

Figure 9: Performance of 1-GNN with with respect to the different activation functions.

representations achieve much lower correlation compared to the case where the model is followed by the sum aggregator. In terms of MSE, 1-GNN, ARMA, ChebNet, 2-hop-GNN and GIN are the best-performing models, while GCN, GAT, GraphSAGE and Provably Powerful GNN are the worst-performing models. For the models trained on PTC-MR, much lower MSE is attained when the models are followed by the sum pooling function. With regards to the three pooling functions, in general, the sum operator outperforms the two other operators. For the models that were trained on

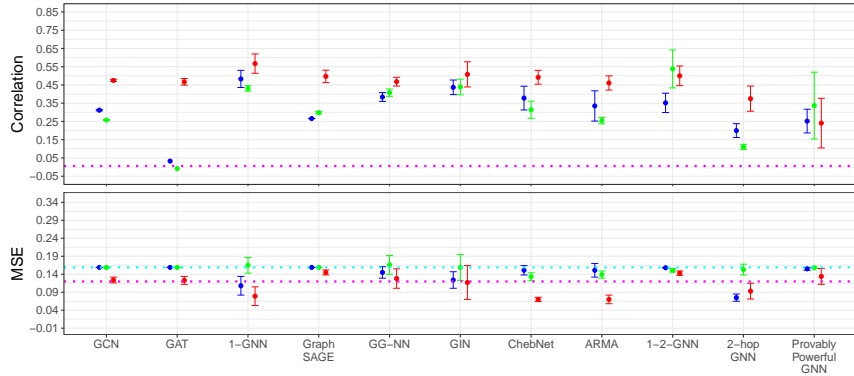

Figure 10: Performance of the different approaches without node features. The GNNs were trained on the MUTAG dataset, and the different colors indicate the three pooling functions: sum (●), mean (●), and max (●). The horizontal lines correspond to the two baselines (random, constant).

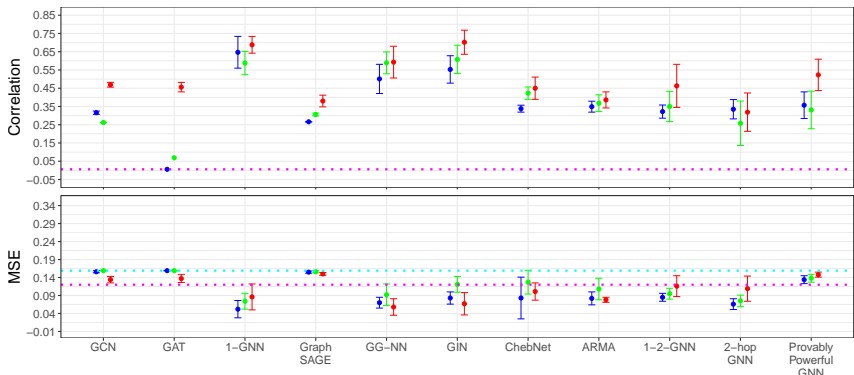

Figure 11: Performance of the different approaches without node features. The GNNs were trained on the PTC-MR dataset, and the different colors indicate the three pooling functions: sum (●), mean (●), and max (●). The horizontal lines correspond to the two baselines (random, constant).

the PTC-MR dataset, the max operator achieves MSEs comparable or lower than those of the sum operator, but also achieves smaller correlations than those of the sum operator in almost all cases.

We next evaluate the expressive power of the different models in the case of graphs whose nodes are annotated with local features such as their degree and the number of triangles in which they participate. The models are first trained on the two aforementioned datasets (i. e., MUTAG and PTC-MR), and then, the synthetic graphs are fed into them to produce a vector representation for each one of those graphs. The results are illustrated in Figures 12 and 13. For both training scenarios, with a single feature, GraphSAGE, GCN, 1-GNN and GAT achieve the highest correlations in general. The lowest MSEs emerge from 1-2-GNN, 1-GNN, 2-hop GNN and GraphSAGE. When both features are passed on to the models, and for models trained on MUTAG, GCN, 1-GNN and ARMA followed by the sum pooling function yield high values of correlation. In terms of MSE, GCN and 1-GNN both with sum pooling and 2-hop GNN with max pooling reach the lower values than the remaining models. For models trained on PTC-MR, Provably Powerful GNN yields the highest correlation with equation 1, followed by Chebnet, ARMA and GCN, in that order. All the models generally achieve the same levels of MSE, but ARMA, 1-2-GNN, 2-hopGNN and Provably Powerful GNN produce slightly lower MSEs than the others. We also observe that the use of the degree as feature leads to an increase in correlation, while the use of an extra feature (i. e., triangle participation) leads to a decrease in correlation. In terms of MSE, we can see that generally the use of a single feature results into a decrease in MSE, while the use of both features leads to a further decrease in MSE. With regards to the pooling functions, in the case of the models trained on MUTAG, the sum operator leads to much better performance (i. e., higher correlations and lower MSEs) compare to the other two operators. In the case of the models trained on PTC-MR, the sum operator still leads to

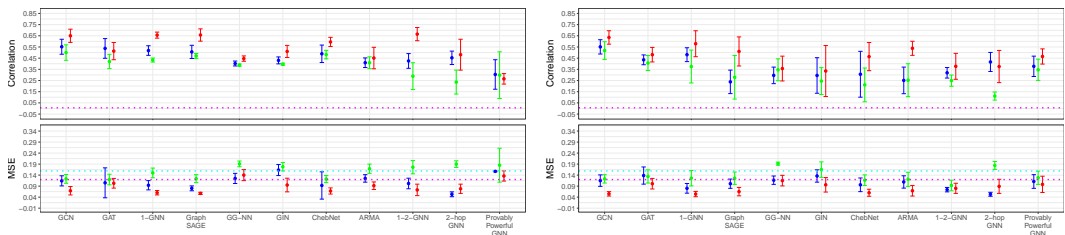

Figure 12: Performance of the different approaches with node features. The GNNs were trained on the MUTAG dataset, and the different colors indicate the three pooling functions: sum (●), mean (●), and max (●). The horizontal lines correspond to the two baselines (random, constant).

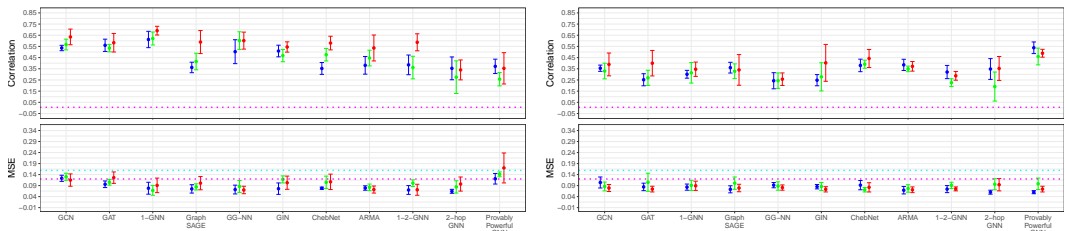

Figure 13: Performance of the different approaches with node features. The GNNs were trained on the PTC-MR dataset, and the different colors indicate the three pooling functions: sum (●), mean (●), and max (●). The horizontal lines correspond to the two baselines (random, constant).

higher correlations in most cases, but in terms of MSE, no pooling function results into consistently better performance than the others.

