# OpenReview forum: "An Empirical Study of the Expressiveness of Graph Kernels and Graph Neural Networks"
_ICLR.cc/2021/Conference — Reject_

### Official Review · AnonReviewer1 · 2020-10-19
**Empirical study on the expressiveness of graph embeddings**

**Rating:** 4
**Confidence:** 5

**Review:**

The paper deals with supervised graph classification comparing GNN and graph kernel approaches. The authors define an intractable graph similarity functions, which boils down to a normalized graph edit distance. The authors then empirically study how well well-known graph kernel and GNN architectures align within their predictions with this distance. To that, the authors generate small-scale, random graph datasets consisting of various types of graphs.

The authors find out that graph kernels still seem to be competitive compared to GNNs, and in some settings, GNNs do not perform better than trivial baselines.

Pro:
- Easy to read
- Cover of related work is good

Cons:
- Most results are already known, and have been found in previous studies
- Not much insight is given, and no methodological contribution (How can we overcome limitations of current models?)
- Generation of synthetic dataset seems arbitrary, no motivation given. Extremely small.

Remarks:
- Please give more motivation for dataset generation and details on generation
- You might consider citing the works of Andreas Loukas on the limits of GNNs
- p. 2. "simialrity" -> similarity
- p. 3. please give a proof/citation that shows that solving (1) is indeed not tractable. Just stating number of permutations as a reason is not enough.
- p. 4. The GR kernel can be trivially extended to work with labeled graphs, moreover RW kernel can also be extended to work with attributed graphs
- p. 4. GraphHopper kernel is not SOTA

Questions:
- Why should the similarity functions of (1) be suitable for all data distributions? E.g., on some data distributions local properties might be more important.
- What is the motivation behind first training on IMDB-BINARY? This seems arbitrary.
- Why did you fix GNN hyperparameters? How were the hyperparameters for the kernels chosen?

---

### Official Review · AnonReviewer4 · 2020-10-28
**A very interesting empirical study, but very hard to interpret.**

**Rating:** 4
**Confidence:** 4

**Review:**

Summary: This paper provides an intensive empirical study on the expressiveness of graph kernels and graph neural networks. It defines an exact but computationally demanding similarity of graphs defined over all possible permutations for adjacency matrices, eq (1), and generate a synthetic dataset of graphs to the scale (up to 9 nodes) where we can compute this quantity. The nomalized kernel value of k_ij of graph kernels and the cosine similarities of GNN-embedded vectors are compared to this exact similarity eq(1) in terms of its correlation coefficients and mean squared errors over carefully chosen graph kernels and graph neural networks in the context of research about the expressiveness. The empirical results are presented when node features are all the same single values (1 or 1.0), and when node features are two values of the degree and the number of triangles in which it participates to. Also, the paper discussed several concrete examples that have a large disagreement on the similarity against the exact similarity eq(1).

Comments: This is a very interesting empirical study including not only GNNs but also graph kernels in the scope, but at the same time, it would be still very hard to interpret those presented results. This study is motivated by many existing papers on the GNN expressivenss, and there should be some explicit efforts (or discussions, at least) to connect and relate these empirical results to what we already know on this theme, for example, isomorphism, WL-test relationship, hard-to-discriminate examples, etc.

- The most confusing points would be the disagreement between the correlation measures and the MSE measures. What can we conclude or learn from these results where graph kernels have lower MSE but also relatively lower correlation than GNNs? The paper even lacks any concluding remarks or conclusion section. (by the way, the caption of Fig.6 should be corrected)

- All of the results are based on the criteria of eq (1), but whether this purely adjacency-matrix based similarity would be worth characterizing the expressiveness aspects of GKs and GNNs would be a bit unconvincing. In particular, we know WL-kernel or several GNNs would have a same level of expressiveness as WL tests. But then, why graph pairs in Figure 6 are hard to get its similarities...? A simple WL test seems enough to discriminate these pairs. Are there any rational explanations for this?

- The pre-training (?) over IMDB-BINARY is also not well motivated, and would lack any context. It's from movie reviews in Internet Movie Database (IMDB). How can we expect any information useful for characterizing the synthetic dataset to be learned?

- Also, the worst case analysis over sum, mean, max aggregators would be also misleading. As seen in the GIN paper (Xu et al, ICLR2019), the expressiveness study suggested that we need sum aggregators (rather than mean or max) for the purpose like the paper's synthetic dataset, didn't it?

- In the context of the expressiveness study, it would be quite of great interest to scrutinize how the exact graph topology relates to the graph kernels or graph neural networks even for very small graphs. But in the practical context, the graph topology is only part of the information to generate a good vector representations of nodes or graphs for downstream tasks. Consider examples like the MNIST graph (28x28 2D grid graph) in the ChebNet paper, and how GNNs can use pixel values representing hand-written characters over the grid.

---

### Official Review · AnonReviewer3 · 2020-10-28
**Interesting analysis but with strong limitations**

**Rating:** 3
**Confidence:** 4

**Review:**

This paper proposes to study the expressiveness of some graph kernels and graph neural networks. To this aim, the authors build a synthetic dataset of graphs and compute the pairwise similarities, which are compared to an oracle function supposed to quantify how much two graphs are isomorphic.

The objective pursued in this paper seems to me to be interesting and useful for practitioners but the manuscript has important limitations.


1- The oracle similarity introduced in the paper has good properties, that one can expect from a similarity function. However, taking this function as a reference oracle in the whole sequel is not properly justified. How is this similarity function canonical? It would be interesting to cite some references on this question.

For example, does the reference oracle give better accuracy scores on some classification problems than the kernels and neural networks used in the paper? This would be an important indication to justify the choice of this reference.

Section 3: Even if I agree that the number of permutation matrices is of order n!, it does not prove that the optimization problem is intractable. There exist other algorithms than exhaustive search.


2- The synthetic dataset is both small (only 191 graphs) and with a small variety of graphs (between 2 and 9 vertices, between 1 and 36 edges). Under these conditions, isn't it difficult to draw solid conclusions?


3- I am not sure that we can expect from a GNN to evaluate accurately the similarity between two graphs from the synthetic dataset at stake, if it was trained on a very different classification dataset?

I have another concern with the fact that some kernels and neural networks capture how the attributes are connected to each other, and do not really catch the topology of an unannotated graph. In this context, giving the same attribute value to all nodes as proposed on page 6 just to be able to use kernels designed for annotated graphs seems strange.


For all these reasons, I believe that the empirical study presented in this paper is not strong enough to justify its publication in this conference.


Typos
page 2: simiarlity
page 2: expessiveness

---

### Official Review · AnonReviewer2 · 2020-10-29
**Empirical expressivity of graph kernels and graph neural networks**

**Rating:** 4
**Confidence:** 4

**Review:**

The paper is an empirical study on the expressive power of graph kernels (GKs) and graph neural networks (GNNs). The paper considers the question whether GKs and GNNs can approximate a fixed graph similarity. The chosen graph similarity is maximal when graphs are isomorphic. Its computation is intractable for most graphs (complexity in $n!$). Experiments are done on very small graphs (at most 9 nodes). Experimental results are provided without any clear conclusion.

One motivation of the paper is to complement many existing studies on the expressive power of GNNs. As most studies consider graph isomorphism testing, the present paper considers graph similarities. But the paper considers an intractable graph similarity related to the graph isomorphism property because the similarity is maximal when the graphs are isomorphic. Therefore it should be made clear what could be the benefit of the proposed approach. Moreover I am quite concerned with the following questions: why should graph kernels correspond to the chosen graph similarity ? Why should the dot product between graph representations at the penultimate layer correspond to the chosen graph similarity ? Experiments are done on very small graphs and the conclusions are not convincing enough. Therefore, in my opinion, the paper is not ready for publication.

Detailed comments.

Introduction. The notion of graph similarity should be made more precise. Many graph similarities could be defined for graphs, some of them only based on the graph structure and others using node or edge features. Here, the paper only considers a structural graph similarity related to graph isomorphism. It is not clear to me whether GKs or GNNs should be able to compute this so-called "true similarity between graphs". It is not clear to me how the ability to compute such a graph similarity is related to the graph isomorphism property.

Related work. Please make precise the relation between this section and Appendix A.1

Section 3. The choice of the similarity measure should be justified. Other graph similarities have been defined in the scientific literature. Some graph similarities use the graph structure and node features. Please make precise the objective of Appendix A.2

Section 4. In Section 4.5.1, a second experiment is considered where "the GNNs are first trained on the IMDB-BINARY dataset". Please explain how are trained the GNNs in the first experiment and in the second experiment. As such I do not see why the dot product between graph representations at the penultimate layer should correspond to the chosen graph similarity. The GNNs could be trained to compute the chosen similarity. I do not understand why this has not been considered.

Section 4. It is not easy to draw conclusions of the experimental results.

The paper does not have a conclusion.

---

### Decision · Program_Chairs · 2021-01-07
**Final Decision**

**Decision:**

Reject

**Comment:**

The reviewers liked the direction of the paper but unanimously agree that, in its current version, it is not strong enough to justify publication at ICLR. There was no rebuttal from the authors to consider.